# MAFLD in Egyptian non-dialysis CKD Patients: Frequency, fibrosis severity, and risk factors

**Zienab M. Saad[1], Hesham K. H. Keryakos[2]\*, Hala A. Hassanin[1], Mahmoud M. Higazi[3], Manar M. Sayed[3], Doaa E. Ismail[4], Safaa M. Abdelhalim[1]**

1 Department of Tropical Medicine, Faculty of Medicine, Minia University, Minya, Egypt, 2 Department of Internal Medicine, Faculty of Medicine, Minia University, Minya, Egypt, 3 Department of Radiology, Faculty of Medicine, Minia University, Minya, Egypt, 4 Department of Clinical Pathology, Faculty of Medicine, Minia University, Minya, Egypt

\* hesham.keryakos@mu.edu.eg

## Abstract

### Introduction

The prevalence of Metabolic Dysfunction-Associated Fatty Liver Disease (MAFLD) in non-dialysis chronic kidney disease (CKD) patients remains underexplored, particularly in high-risk populations such as Egyptians with high rates of metabolic disorders.

### Aims

This study aimed to determine the frequency of MAFLD in Egyptian non-dialysis CKD patients, assess liver fibrosis severity, and identify associated metabolic risk factors.

### Methods

A cross-sectional study of 108 CKD patients was conducted, with participants stratified into MAFLD (n=64) and non-MAFLD (n=44) groups. Diagnostic criteria for MAFLD included hepatic steatosis (ultrasonography) and metabolic risk factors. Non-invasive fibrosis scores (NAFLD score, FIB-4 index, APRI score) and shear wave elastography were used to evaluate liver fibrosis.

### Results

MAFLD prevalence was 59.25%. MAFLD patients exhibited significant associations with older age (56±17.1 vs. 43±17.1 years, *p<0.001*), higher BMI (34.5±6.2 vs. 27.2±5.7 kg/m², *p<0.001*), diabetes (48.4% vs. 4.5%, *p<0.001*), hypertension (68.7% vs. 22.7%, *p<0.001*), and insulin resistance (HOMA-IR: 3.5±3.6 vs. 1.45±1.03, *p<0.001*). MAFLD patients also had lower eGFR (40.5±28.0 vs. 58.9±39.9 mL/min/1.73 m², *p=0.017*) and higher liver stiffness (7.6±1.8 vs. 6.9±1.7 kPa, *p=0.047*), with advanced fibrosis more prevalent in later CKD stages..

**Data availability statement:** The data underlying the findings of this study are freely available in Zenodo at https://doi.org/10.5281/zenodo.17144136.

**Funding:** The author(s) received no specific funding for this work.

**Competing interests:** The authors have declared that no competing interests exist.

## Conclusion

MAFLD is highly prevalent among non-dialysis CKD patients, driven by shared metabolic abnormalities and CKD severity. These findings highlight the bidirectional relationship between MAFLD and CKD, emphasizing the need for integrated screening and management strategies to mitigate progression risks in this population.

## Introduction

The liver plays a central role in metabolic homeostasis, and chronic liver disease can lead to severe systemic complications. Among these, **Metabolic Dysfunction-Associated Fatty Liver Disease (MAFLD)** is the most prevalent chronic liver disorder worldwide, affecting **20–30% of the global adult population** [1,2]. Unlike the previous term **nonalcoholic fatty liver disease (NAFLD)**, which was exclusionary and did not account for the multisystem nature of metabolic dysfunction, the revised **MAFLD criteria** explicitly link hepatic steatosis (≥5% fat accumulation) defined histologically or by non-invasive surrogates such as ultrasound, controlled attenuation parameter (CAP), or magnetic resonance imaging–proton density fat fraction (MRI-PDFF) to underlying metabolic abnormalities, including **obesity, insulin resistance, type 2 diabetes mellitus (T2DM), hypertension, and dyslipidemia** [3,4]. This shift in nomenclature reflects the growing recognition of MAFLD as a **metabolic liver disorder with systemic consequences**, rather than merely a hepatic condition.

The clinical spectrum of MAFLD ranges from **simple steatosis** to **fibrosis, cirrhosis, and hepatocellular carcinoma** [3]. Although "Metabolic dysfunction-associated steatotic liver disease (MASLD)" has been proposed as the updated nomenclature in 2023, we adopted the MAFLD framework throughout this study, as it reflects the diagnostic criteria used during recruitment and remains widely employed in nephrology-focused research. Alarmingly, MAFLD is now the **fastest-growing indication for liver transplantation** in the U.S. [1], underscoring its escalating burden. Furthermore, MAFLD frequently coexists with **chronic kidney disease (CKD)**, another major public health concern affecting **over 25% of adults aged ≥65 years** [5,6]. Both conditions share common pathogenic pathways, including **chronic inflammation, oxidative stress, lipotoxicity, and insulin resistance** [5]. Prior studies show conflicting evidence regarding the relationship between MAFLD and CKD. Some report MAFLD independently increases CKD incidence and fibrosis severity [7,8], while others suggest no independent effect after adjusting for obesity and diabetes [9], leaving the nature of this association unresolved [4].

In Egypt, the prevalence of metabolic risk factors is among the highest globally, with obesity affecting ~36% of adults [10], type 2 diabetes affecting ~22% [11,12], and CKD prevalence around 15% [6,13]. These statistics underscore the need to investigate the interplay between MAFLD and CKD in this high-risk population. However, data on MAFLD frequency in **non-dialysis CKD patients remain scarce**, particularly in Middle Eastern populations. Therefore, this study aimed to determine

the prevalence of MAFLD in Egyptian non-dialysis CKD patients, identify shared metabolic risk factors, and assess the relationship between CKD severity and liver fibrosis progression.

These findings may inform **early screening and targeted interventions** to mitigate the dual burden of MAFLD and CKD in high-risk populations

## Subjects and methods

### Study design and participants

This was a cross-sectional observational study conducted **at Minia University Hospital** between March and September 2023. A total of **108 non-dialysis adult patients with CKD** were consecutively recruited from the **Nephrology Department**. Of these, **64 patients (59.25%)** met the diagnostic criteria for MAFLD.

The study included adults aged >18 years with CKD of any etiology (excluding obstructive uropathy), while exclusion criteria comprised CKD secondary to obstructive uropathy, end-stage kidney disease (ESKD) patients on dialysis, history or evidence of other chronic liver diseases (e.g., autoimmune hepatitis, cirrhosis, or hepatocellular carcinoma), positive viral hepatitis markers (HBsAg or anti-HCV), or significant alcohol intake. Although MAFLD does not require exclusion of alcohol, in our Egyptian cohort alcohol consumption is rare; therefore, to avoid diagnostic ambiguity we excluded patients with any alcohol use. Non-dialysis CKD stage G5 patients (n = 10) were included.

### Ethics

The study protocol was reviewed and approved by the Institutional Review Board of the Faculty of medicine, Minia University (Approval No. 571/2023). All procedures performed in the study were conducted in accordance with the ethical standards of the institutional research committee and with the Declaration of Helsinki. Written informed consent was obtained from all participants prior to enrollment.

### Diagnostic criteria of CKD

CKD was diagnosed based on either kidney damage markers or reduced kidney function (i.e., estimated glomerular filtration rate [eGFR] <60 ml/min/1.73 m$^2$ body surface area) persisting for ≥3 months. Kidney damage indicators included urine albumin-to-creatinine ratio [UACR] >30 mg/g, active urine sediment, tubular dysfunction-related electrolyte abnormalities, kidney transplant history, or structural abnormalities on biopsy/imaging.

Patients were then classified into stages based on the severity of the disease defined by the level of eGFR and the degree of albuminuria. Disease staging followed KDIGO guidelines, categorizing patients by eGFR (G1(normal or high): ≥90; G2 (mildly increased): 60–89; G3a (mildly to moderately increased): 45–59; G3b (moderately to severely increased): 30–44; G4 (severely increased): 15–29; G5: <15 mL/min/1.73 m$^2$) and albuminuria (A1(normal to mildly increased): <30; A2 (moderately increased): 30–299; A3 (severely increased): ≥300 mg/g creatinine). Patients with G3b-G5 or A3 were classified as high/very high risk according to current guidelines [14].

### Diagnostic criteria of MAFLD

MAFLD diagnosis required hepatic steatosis documented either by non-invasive biomarkers (APRI score, FIB-4 index, NAFLD fibrosis score) or imaging (ultrasonography, shear wave elastography) in individuals meeting one of three criteria: [1] overweight/obesity (BMI ≥ 25 kg/m$^2$ for overweight and ≥30 kg/m$^2$ for obesity using Middle East/North Africa criteria) [15]; [2] type 2 diabetes mellitus; or [3] metabolic dysregulation. Metabolic dysregulation was defined in lean/normal-weight individuals (BMI < 25 kg/m$^2$) as having ≥2 of the following: elevated waist circumference (≥94 cm for men and ≥80 cm for women), blood pressure ≥130/85 mmHg or antihypertensive treatment, triglycerides ≥150 mg/dL (1.70 mmol/L) or treatment, reduced HDL-C (<40 mg/dL [1.0 mmol/L] men/<50 mg/dL [1.3 mmol/L] women), prediabetes (fasting glucose 100–125 mg/

dL [5.6–6.9 mmol/L], 2-hour post-load glucose 140–199 mg/dL [7.8–11.0 mmol/L], or HbA1c 5.7–6.4% [39–47 mmol/mol]), HOMA-IR ≥ 2.5, or hs-CRP > 2 mg/L [3].

## Clinical data collection and laboratory methods

Comprehensive demographic, anthropometric, and clinical data were collected at enrollment, including height, weight, BMI (calculated as weight[kg]/height[m$^2$]), and waist circumference (measured at the midpoint between the 12th rib and iliac crest). BMI classifications were: normal (20.0–24.9 kg/m$^2$), overweight (25.0–29.9 kg/m$^2$), and obese (≥30.0 kg/m$^2$). Fasting blood samples were analyzed for complete blood count (Sysmex KX-21N, Japan), coagulation profile (STAGO COMPACT CT, USA), and biochemical parameters (Konelab 20i, Finland), including liver/renal function, glucose metabolism, and lipid profile. Renal function was assessed using creatinine-based eGFR equations (Cockcroft-Gault, MDRD, CKD-EPI) and spot urine albumin-to-creatinine ratio (UACR) using the second morning void.

Insulin resistance was evaluated using homeostasis model assessment IR index (HOMA-IR): (fasting insulin [μIU/mL] × fasting glucose [mg/dL])/ 405, with values <1.0 indicating normal sensitivity, > 1.9 suggesting early resistance, and >2.9 indicating significant resistance.

Liver fibrosis was assessed through both biomarker panels and imaging methods: the APRI score [(AST/ULN) × 100/ platelets (10⁹/L]) with cutoffs of <0.5 (no fibrosis), > 0.5 (fibrosis), and >1.5 (probable cirrhosis); FIB-4 index [Age × AST/ (platelets × √ALT)] with cutoffs of <1.45 (no fibrosis), 1.45–3.25 (mild-moderate fibrosis, F1-F2), and >3.25 (advanced fibrosis/cirrhosis, F3-F4); and the NAFLD fibrosis score (incorporating age, BMI, diabetes status, AST/ALT ratio, platelets, and albumin) with cutoffs of <−1.455 (F0-F2), −1.455-0.675 (indeterminate), and >0.675 (F3-F4). Imaging evaluation included ultrasonographic assessment of hepatic steatosis through liver echogenicity, vascular wall visibility, and diaphragmatic outline, along with 2D-Shear Wave Elastography (GE Logiq E9, convex probe) performed on the right hepatic lobe (3–5 cm depth, avoiding vascular structures) during breath-holding, where five valid measurements (demonstrating homogeneous color mapping, parallel propagation lines, and IQR/median <30%) were obtained and averaged for the final liver stiffness quantification.

## Statistical analysis

Data were coded, tabulated, and analyzed using IBM SPSS Statistics (Version 24, IBM Corp., USA). Continuous parametric variables were expressed as mean ± standard deviation (SD), while non-parametric data were reported as median (range). Categorical variables were summarized as numbers and percentages (n, %). Comparisons between parametric continuous variables were performed using the independent t-test, whereas the Mann-Whitney U test was used for non-parametric data. Categorical variables were analyzed using the Chi-square ($\chi^2$) test or Fisher's exact test, as appropriate. Correlations between variables were assessed using Spearman's rank correlation coefficient.

We modeled MAFLD status using multivariable logistic regression with deliberately non-redundant covariate sets to avoid overlapping constructs. Continuous predictors (age, BMI, HOMA-IR, and HbA1c) were centered and standardized (z-scores) prior to modeling to improve numerical stability and interpretability. We prespecified four models: (i) Model 1 included age, sex, BMI (z), HOMA-IR (z), diabetes mellitus (DM), and hypertension (HTN) but excluded HbA1c; (ii) Model 2 swapped DM for HbA1c (z) while retaining HTN; (iii) Model 3 included age, sex, BMI (z), HOMA-IR (z), and DM only; and (iv) Model 4 included age, sex, BMI (z), HOMA-IR (z), and HbA1c (z) only. Model fit was summarized by Akaike Information Criterion (AIC) for standard logistic models. To evaluate potential multicollinearity among correlated metabolic covariates (BMI, DM, HTN, HOMA-IR, HbA1c), we calculated variance inflation factors (VIFs) for the final specifications; values <3 were taken to indicate no problematic multicollinearity. Across all retained models, VIFs were below commonly accepted thresholds (S2 Table). To guard against small-sample bias and potential separation in strata, we performed Firth bias-reduced logistic regression sensitivity analyses using the same covariate sets; the pattern and magnitude of effects were concordant with the standard models (Supplementary S1, S3, and S4Tables).

Diagnostic performance was evaluated via receiver operating characteristic (ROC) curve analysis, determining the area under the curve (AUC), optimal cutoff value, sensitivity, positive predictive value (PPV), negative predictive value (NPV), and accuracy. Statistical significance was set at $p \leq 0.05$, with $p \leq 0.01$ considered highly significant and $p \leq 0.001$ very highly significant.

## Results

### Demographic, anthropometric and laboratory characteristics

Comparative analysis revealed significant differences between CKD patients with and without MAFLD, highlighting a distinct metabolic-renal phenotype (Table 1). MAFLD patients were notably older ($56 \pm 17.1$ vs. $43 \pm 17.1$ years, **p<0.001**) and more likely to be female (46.9% vs. 22.7%, **p=0.011**), suggesting potential age-related metabolic dysfunction and sex-specific risk factors in disease development. No significant difference in smoking status (21.9% vs. 29.5%, p=0.366).

These patients exhibited substantially worse metabolic profiles, with higher BMI ($34.5 \pm 6.2$ vs. $27.2 \pm 5.7$ kg/m$^2$, **p<0.001**), greater waist circumference (males: $96.1 \pm 11.7$ vs. $81.7 \pm 11.4$ cm; females: $105 \pm 9.57$ vs. $95.4 \pm 7.9$ cm; **p<0.01** for both), and dramatically higher prevalence of diabetes (48.4% vs. 4.5%, **p<0.001**) and hypertension (68.7% vs. 22.7**%, p<0.001**). These findings underscore the clustering of metabolic syndrome components in MAFLD-CKD patients, with central obesity emerging as a particularly strong driver of hepatic steatosis in this population.

Medication patterns revealed important clinical insights, with MAFLD patients showing significantly higher use of RAAS inhibitors (58.2% vs. 42.1%, p=0.032), insulin therapy (32.8% vs. 9.5%, **p<0.001**), and diuretics (39.1% vs. 21.4%, p=0.027). This pattern reflects both the greater disease severity in this group and current treatment approaches, while also suggesting that standard therapies may be insufficient to prevent MAFLD progression in CKD.

Laboratory results further delineate the MAFLD phenotype, showing evidence of liver injury (elevated ALT/AST and bilirubin), worse renal function (lower eGFR) (Fig 1), higher uric acid, and prominent metabolic derangements including higher fasting glucose ($97.2 \pm 24.9$ vs. $85.2 \pm 16.6$ mg/dL, **p=0.006**), higher HbA1c ($6.2 \pm 1.5$% vs. $5.2 \pm 0.4$%, **p<0.001**), increased insulin resistance (HOMA-IR $3.5 \pm 3.6$ vs. $1.45 \pm 1.03$, **p<0.001**), and dyslipidemia (higher triglycerides: $161.9 \pm 62.1$ vs. $105 \pm 48.7$ mg/dL, **p<0.001** and LDL: $95.3 \pm 27.8$ vs. $83.8 \pm 8$ mg/dL, **p=0.037**). These abnormalities suggest interconnected pathophysiological mechanisms involving chronic inflammation, oxidative stress, and lipid metabolism disorders that simultaneously affect both liver and kidney function.

### Liver fibrosis assessment

**Non-invasive scores.** Comparative analysis of non-invasive liver fibrosis scores revealed significant difference between MAFLD and non-MAFLD patients (Table 2). The **NAFLD Fibrosis Score (NFS)** was markedly elevated in MAFLD patients ($0.96 \pm 0.89$ vs. $0.44 \pm 0.61$, **p<0.001**), suggesting greater fibrosis risk in this group. In contrast, the **FIB-4 index** showed no significant difference between groups ($1.23 \pm 0.55$ vs. $1.14 \pm 0.93$, p=0.558). Interestingly, the **APRI score** demonstrated an inverse trend, being lower in MAFLD patients ($0.27 \pm 0.15$ vs. $0.39 \pm 0.51$, p=0.093), with both groups below the 0.5 fibrosis threshold. The wide variability in non-MAFLD APRI scores (SD $\pm 0.51$) indicates limited specificity of this marker for metabolic liver disease.

### Shear wave elastography

Shear wave elastography revealed significant differences in liver stiffness between MAFLD and non-MAFLD patients. The MAFLD group exhibited higher mean stiffness values ($7.6 \pm 1.8$ kPa vs $6.9 \pm 1.7$ kPa, p=0.047) with broader interquartile ranges (6.2–8.9 kPa vs 5.8–7.8 kPa), indicating greater fibrotic burden (Table 3). This pattern corresponded with distinct

**Table 1. Demographic and Clinical Characteristics of Study Participants.**

| Variable | MAFLD Group (n = 64, 59.25%) | Non-MAFLD Group (n = 44, 40.7%) | *p*-value |
|---|---|---|---|
| **Demographics** | | | |
| Age (years) | 56 ± 17.1 (28-81) | 43 ± 17.1 (21-82) | <0.001* |
| Female sex (%) | 46.9% | 22.7% | 0.011* |
| Current smokers (%) | 21.9% | 29.5% | 0.366 |
| **Anthropometrics** | | | |
| BMI (kg/m²) | 34.5 ± 6.2 (24-53) | 27.2 ± 5.7 (19-42) | <0.001* |
| Waist circumference (cm) | | | |
| - Male | 96.1 ± 11.7 (70-120) | 81.7 ± 11.4 (60-109) | <0.001* |
| - Female | 105 ± 9.57 (90-120) | 95.4 ± 7.9 (80-109) | 0.006* |
| **Comorbidities** | | | |
| Diabetes mellitus (%) | 48.4% | 4.5% | <0.001* |
| Hypertension (%) | 68.7% | 22.7% | <0.001* |
| Medication Use (%) | | | |
| -RAAS inhibitors | 58.2% | 42.1% | 0.032* |
| -Statins | 45.3% | 28.6% | 0.041* |
| -Insulin therapy | 32.8% | 9.5% | <0.001* |
| -Diuretics | 39.1% | 21.4% | 0.027* |
| **Hematology** | | | |
| Hemoglobin (g/dL) | 10.95 ± 2.17 (8–14) | 11.4 ± 2.3 (7–14) | 0.070 |
| Platelets (10³/µL) | 249.8 ± 77.3 (108-350) | 244.7 ± 75.5 (110-350) | 0.735 |
| **Liver Function** | | | |
| ALT (U/L) | 32.5 ± 19.2 (10-113) | 24.2 ± 13.3 (10-71) | 0.013* |
| AST (U/L) | 37 ± 22 (10-145) | 28.4 ± 16.6 (10-71) | 0.029* |
| Total bilirubin (mg/dL) | 0.87 ± 0.32 (0.3-2.5) | 0.66 ± 0.48 (0.2-2.5) | 0.008* |
| Direct bilirubin (mg/dL) | 0.42 ± 0.22 (0.1-1.1) | 0.29 ± 0.22 (0.09-1.1) | 0.002* |
| **Renal Function** | | | |
| eGFR (mL/min/1.73 m²) | 40.5 ± 28.0 (5.8-119.3) | 58.9 ± 39.9 (12.8-135.1) | 0.017* |
| Uric acid (mg/dL) | 6.5 ± 2.8 (2.8-14.5) | 4.6 ± 1.5 (2.8-10.1) | 0.001* |
| **Metabolic Parameters** | | | |
| Fasting glucose (mg/dL) | 97.2 ± 24.9 (60-160) | 85.2 ± 16.6 (61-119) | 0.006* |
| HbA1c (%) | 6.2 ± 1.5 (4.5-12) | 5.2 ± 0.4 (4.5-6.5) | <0.001* |
| HOMA-IR | 3.5 ± 3.6 (0.8-16.5) | 1.45 ± 1.03 (0.3-4.2) | <0.001* |
| **Lipid Profile** | | | |
| Cholesterol (mg/dL) | 178.6 ± 50 (100-300) | 140.8 ± 38.7 (76-219) | <0.001* |
| Triglycerides (mg/dL) | 161.9 ± 62.1 (73-350) | 105 ± 48.7 (51-335) | <0.001* |
| LDL (mg/dL) | 95.3 ± 27.8 (48-165) | 83.8 ± 8 (36-161) | 0.037* |

BMI = Body Mass Index; ALT = Alanine Aminotransferase; AST = Aspartate Aminotransferase; eGFR = estimated Glomerular Filtration Rate; HOMA-IR = Homeostatic Model Assessment of Insulin Resistance; LDL = Low-Density Lipoprotein

Data presented as mean ± standard deviation (range) or percentage

Medication use was self-reported and verified via medical records.

Statistically significant differences (p < 0.05) marked with *

Group comparisons use independent t-tests for continuous variables and chi-square tests for categorical variables

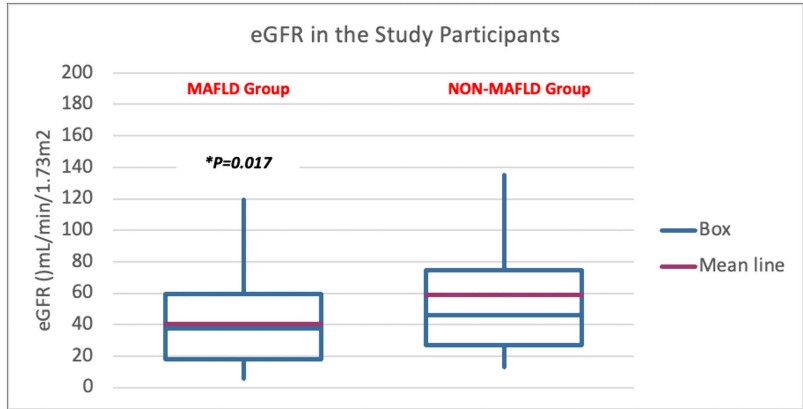

**Fig 1. eGFR in the study participants.**

**Table 2. Comparison of Non-invasive Liver Fibrosis Scores.**

| Score | MAFLD (n=64) | Non-MAFLD (n=44) | p-value |
|---|---|---|---|
| NAFLD Fibrosis Score | 0.96±0.89 | 0.44±0.61 | **<0.001*** |
| FIB-4 Index | 1.23±0.55 | 1.14±0.93 | 0.558 |
| APRI Score | 0.27±0.15 | 0.39±0.51 | 0.093 |

MAFLD=metabolic dysfunction-associated fatty liver disease; NAFLD=non-alcoholic fatty liver disease; FIB-4=Fibrosis-4 index; APRI=AST-to-platelet ratio index.

Data presented as mean±standard deviation.

Statistical significance set at **p<0.05** (marked with *).

Group comparisons performed using independent t-tests for normally distributed data.

**Table 3. Liver Stiffness and Fibrosis Grading by MAFLD Status.**

| Parameter | MAFLD Group (n=64) | Non-MAFLD Group (n=44) | p-value |
|---|---|---|---|
| **Liver stiffness (kPa)** | | | 0.047* |
| -Range | 4.2-12 | 4.2-10 | |
| -Mean±SD | 7.6±1.8 | 6.9±1.7 | |
| -IQR | 6.2–8.9 | 5.8–7.8 | |
| **Fibrosis Grade** | n (%) | n (%) | 0.017* |
| - F0 (None) | 11 (17.2) | 12 (27.3) | |
| - F1 (Mild) | 12 (18.8) | 16 (36.4) | |
| - F2 (Moderate) | 41 (64.1) | 16 (36.4) | |

Measurement success rate: 92% valid measurements (IQR/median <30%)

Fibrosis staged per METAVIR criteria: F0 (no fibrosis), F1 (portal fibrosis without septa), F2 (few septa).

Stiffness cutoffs: F0-F1 (<7.0 kPa), F2 (7.0–9.5 kPa), F3-F4 (>9.5 kPa) based on manufacturer's reference ranges.

IQR=Interquartile range (25th-75th percentile).

p-values from Mann-Whitney U test (stiffness) or χ² test (fibrosis grades).

fibrosis distributions, as 64.1% of MAFLD patients showed moderate fibrosis (F2) compared to only 36.4% of non-MAFLD cases. Conversely, non-MAFLD patients had higher proportions of normal liver architecture (F0: 27.3% vs 17.2%) and mild fibrosis (F1: 36.4% vs 18.8%, p=0.017) (**Table 3**, **Fig 2**).

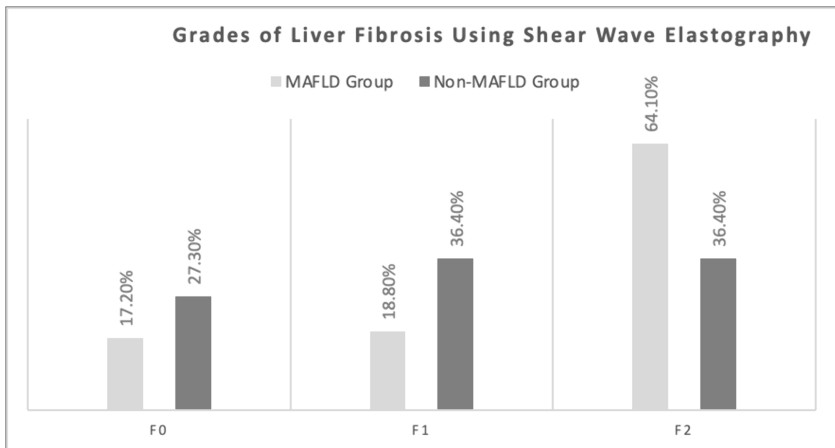

**Fig 2. Grades of liver fibrosis in MAFLD group (grey) and non-MAFLD group (black).**

The narrower IQR in non-MAFLD patients (2.0 kPa vs 2.7 kPa) suggests more consistent stiffness measurements, while the wider dispersion in MAFLD may reflect variable individual susceptibility to metabolic injury. The consistent statistical significance ($p < 0.05$) across both continuous stiffness measurements and categorical fibrosis grades strengthens the evidence for MAFLD-specific fibrotic progression. These findings demonstrate that MAFLD in CKD patients is associated with both quantitatively and qualitatively different patterns of liver fibrosis compared to non-MAFLD counterparts.

### CKD stage and liver fibrosis

The analysis revealed significant differences in CKD progression between patients with and without MAFLD (**Fig 3**). MAFLD patients demonstrated more advanced renal disease, with a 40.7% prevalence of late-stage CKD (G4-G5) compared to 31.8% in non-MAFLD cases. Most strikingly, kidney failure (G5) was nearly five times more common in the MAFLD group (21.9% vs 4.5%, OR=4.9), suggesting MAFLD may accelerate progression to end-stage renal disease. In contrast, non-MAFLD patients showed predominance of earlier CKD stages (G1-G2: 40.9% vs 25.1%), with G1 (normal/ high eGFR with kidney damage) being 3.6 times more prevalent (22.7% vs 6.3%), potentially reflecting different underlying etiologies such as glomerulonephritis or vascular disease.

These findings demonstrate a clear association between MAFLD and advanced CKD, particularly kidney failure, while non-MAFLD patients predominantly had earlier disease stages. The 34.4% prevalence of moderate CKD (G3) in MAFLD patients versus 27.35% in non-MAFLD further reinforces the link between metabolic liver disease and renal dysfunction progression. This stage-specific distribution underscores the clinical importance of integrated metabolic-liver-renal management in MAFLD patients, with particular attention needed for those showing concurrent liver fibrosis and renal impairment.

### Association Between CKD Progression and Liver Fibrosis in MAFLD Patients

Our analysis demonstrated a significant relationship between CKD stage and liver fibrosis severity in MAFLD patients (**Table 4**, **Fig 4**). Patients with late-stage CKD (eGFR < 60 mL/min/1.73 m$^2$) showed substantially higher liver stiffness values compared to early-stage counterparts (7.9 ± 1.8 kPa vs 6.6 ± 1.69 kPa; $p = 0.013$). While fibrosis grade distribution did not reach statistical significance ($p = 0.394$), clinically relevant patterns emerged: late-stage CKD patients had a 38% higher prevalence of moderate fibrosis (F2: 68.8% vs 50%) and nearly double the proportion of advanced fibrosis (F3-F4:

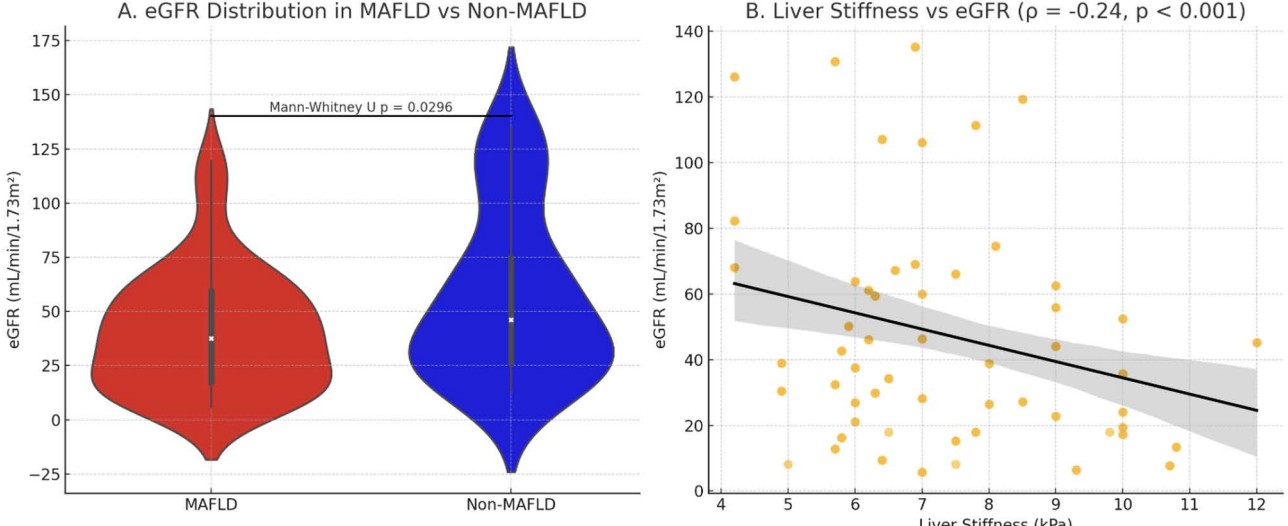

**Fig 3. A. Violin plots showing eGFR distribution in MAFLD (red) vs. non-MAFLD (blue) groups.** Central lines: medians; boxes: IQR; whiskers: 1.5×IQR. B. Correlation between liver stiffness and eGFR (r=−0.62, *p*<0.001). Shaded area: 95% CI.

**Table 4. Liver Fibrosis Characteristics by CKD Stage in MAFLD Patients (n=64).**

| Variable | Early stage CKD (n=16) (eGFR≥60 ml/min/1.73 m²) | Late stage CKD (n=48) (eGFR<60 ml/min/1.73 m²) | *p*-value |
|---|---|---|---|
| Liver stiffness (kPa) | 6.6±1.69 | 7.9±1.8 | 0.013* |
| Grades of fibrosis | | | 0.394 |
| F0 (None) | 4 (25%) | 7 (14.6%) | |
| F1 (Mild) | 4 (25%) | 8 (16.7%) | |
| F2 (Moderate) | 8 (50%) | 33 (68.8%) | |

Independent sample t test for quantitative data between groups, Chi square test for qualitative data between groups.

*Significant level at p value≤0.05.

14.6% vs 25%). These findings suggest that declining renal function may exacerbate hepatic fibrogenesis in MAFLD, though the small early-stage subgroup (n=16) may limit statistical power for categorical analyses.

## Multivariate analysis of MAFLD predictors in CKD patients

To identify independent predictors of MAFLD in our CKD cohort, we performed multivariable logistic regression with several pre-specified models to mitigate multicollinearity among correlated metabolic variables (e.g., diabetes mellitus, HbA1c, hypertension). The results were confirmed by sensitivity analysis using Firth's bias-reduced method (Table 5).

Across all models, a higher body mass index (BMI) was consistently and strongly associated with an increased risk of MAFLD. For instance, in Model 4 (which included HbA1c and HOMA-IR but not diabetes or hypertension as separate variables), a one-standard-deviation increase in BMI was associated with an approximately 5.7-fold higher odds of MAFLD (OR 5.69, 95% CI 1.95–16.61, *p=0.001*). Measures of glycemic control and insulin resistance were also significant independent predictors. Both HbA1c and HOMA-IR emerged as strong risk factors in models that did not include the other. In Model 4, the odds of MAFLD increased approximately 9.4-fold per standard deviation increase in HbA1c (OR 9.35,

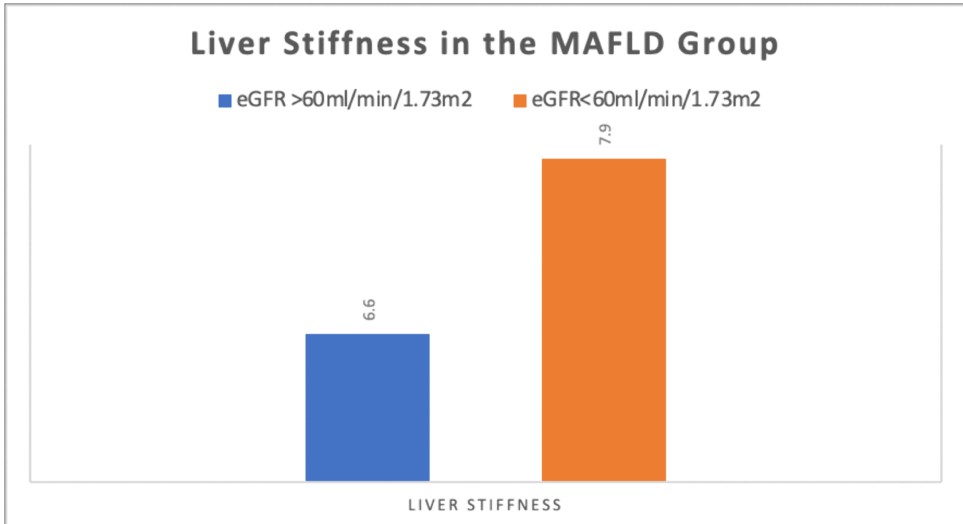

**Fig 4. Stratification of Liver Stiffness according to eGFR in the MAFLD group.**

95% CI 2.33–37.47, *p = 0.002*) and 8.7-fold per standard deviation increase in HOMA-IR (OR 8.70, 95% CI 1.31–57.74, *p = 0.025*).

The association of clinical diagnoses like diabetes and hypertension with MAFLD depended on the model specification. When modeled without their biochemical surrogates (Model 1), hypertension was a significant risk factor (OR 5.15, 95% CI 1.63–16.26, p = 0.005), while diabetes mellitus showed a strong but non-significant trend (OR 5.22, 95% CI 0.87–31.33, p = 0.071). However, when HbA1c was included in the model (Model 2), the association for hypertension was attenuated and non-significant (OR 2.88, 95% CI 0.82–10.09, p = 0.099), suggesting that the risk conveyed by hypertension is mediated, in part, by its association with hyperglycemia. Age and sex were not statistically significant predictors in any model. In summary, this analysis identifies elevated BMI, insulin resistance (HOMA-IR), and poor glycemic control (HbA1c) as the dominant, independent metabolic risk factors for MAFLD in patients with CKD.

### After medications adjustment

After adjusting for medication effects (Table 6), several associations were modified. HbA1c remains the strongest predictor though less extreme, emphasizing residual hyperglycemia risk beyond medication effects. HOMA-IR became borderline significant, implying insulin resistance's impact is largely treatable with current therapies. These results emphasize the importance of considering medication effects when evaluating metabolic risk factors, while confirming the central role of glycemic control in MAFLD pathogenesis among CKD patients.

### ROC curve Analysis

The ROC analysis demonstrates that the NAFLD Fibrosis Score (AUC = 0.82) outperforms both FIB-4 (AUC = 0.71) and APRI (AUC = 0.65) in detecting liver fibrosis among MAFLD patients with CKD (**Fig 5**). Its superior performance likely stems from its incorporation of metabolic parameters (BMI, diabetes status) and liver-specific markers that better reflect the pathophysiology of MAFLD, particularly in the context of renal impairment. The score's strong discriminatory power suggests it should be the preferred non-invasive tool for fibrosis risk stratification in this population, especially when liver biopsy is contraindicated or unavailable.

**Table 5. Multivariable logistic regression models for MAFLD predictors.**

| Model Specification | Variable | Standard Logistic Regressions | | Firth Penalized Logistic Regression | |
|---|---|---|---|---|---|
| | | OR (95% CI) | *p*-value | OR (95% CI) | *p*-value |
| **Model 1:** **DM & HTN** (AIC = 93.70) | Age (z-score) | 0.72 (0.36–1.43) | 0.348 | 0.78 (0.41–1.49) | 0.452 |
| | Male Sex | 2.99 (0.60–14.92) | 0.181 | 2.66 (0.58–12.24) | 0.209 |
| | **BMI (z-score)** | **7.82 (2.52–24.25)** | **<0.001** | **6.08 (2.14–17.30)** | **<0.001** |
| | HOMA-IR (z-score) | 3.42 (0.62–18.89) | 0.159 | 2.46 (0.57–10.70) | 0.229 |
| | Diabetes Mellitus | 5.22 (0.87–31.33) | 0.071 | 3.70 (0.75–18.32) | 0.109 |
| | **Hypertension** | **5.15 (1.63–16.26)** | **0.005** | **4.59 (1.54–13.65)** | **0.006** |
| **Model 2:** **HbA1c & HTN** (AIC = 91.66) | Age (z-score) | 0.70 (0.36–1.39) | 0.311 | 0.74 (0.39–1.43) | 0.373 |
| | Male Sex | 1.45 (0.27–7.69) | 0.661 | 1.41 (0.29–6.83) | 0.671 |
| | **BMI (z-score)** | **6.25 (2.05–19.12)** | **0.001** | **5.04 (1.77–14.38)** | **0.002** |
| | HOMA-IR (z-score) | 5.48 (0.82–36.43) | 0.078 | 4.22 (0.73–24.28) | 0.106 |
| | Hypertension | 2.88 (0.82–10.09) | 0.099 | 2.66 (0.80–8.81) | 0.110 |
| | **HbA1c (z-score)** | **5.83 (1.28–26.49)** | **0.022** | **4.60 (1.13–18.68)** | **0.033** |
| **Model 3:** **DM only** (AIC = 99.96) | Age (z-score) | 0.84 (0.43–1.61) | 0.591 | 0.87 (0.47–1.64) | 0.675 |
| | Male Sex | 3.37 (0.74–15.44) | 0.118 | 3.01 (0.69–13.02) | 0.141 |
| | **BMI (z-score)** | **7.73 (2.65–22.52)** | **<0.001** | **6.32 (2.30–17.40)** | **<0.001** |
| | HOMA-IR (z-score) | 4.77 (0.83–27.45) | 0.081 | 4.04 (0.78–21.01) | 0.097 |
| | **Diabetes Mellitus** | **6.56 (1.12–38.57)** | **0.037** | 4.80 (0.95–24.19) | 0.057 |
| **Model 4:** **HbA1c only** (AIC = 92.41) | Age (z-score) | 0.72 (0.37–1.40) | 0.334 | 0.75 (0.39–1.43) | 0.381 |
| | Male Sex | 1.19 (0.24–5.95) | 0.831 | 1.16 (0.25–5.45) | 0.847 |
| | **BMI (z-score)** | **5.69 (1.95–16.61)** | **0.001** | **4.78 (1.73–13.25)** | **0.003** |
| | **HOMA-IR (z-score)** | **8.70 (1.31–57.74)** | **0.025** | **7.16 (1.18–43.42)** | **0.032** |
| | **HbA1c (z-score)** | **9.35 (2.33–37.47)** | **0.002** | **7.40 (2.01–27.16)** | **0.003** |

Abbreviations: OR, Odds Ratio; CI, Confidence Interval; AIC, Akaike Information Criterion; DM, Diabetes Mellitus; HTN, Hypertension.

*Note: All continuous variables (Age, BMI, HOMA-IR, HbA1c) were standardized (z-scores) prior to analysis. Statistically significant results (p < 0.05) are in bold.*

**Table 6. Medication-inclusive multivariable logistic regression for MAFLD.**

| Variable | Adjusted OR (95% CI) | p-value |
|---|---|---|
| Age (z-score) | 0.70 (0.36–1.39) | 0.311 |
| Male Sex | 1.45 (0.27–7.69) | 0.661 |
| **BMI (z-score)** | **6.25 (2.05–19.12)** | **0.001*** |
| HOMA-IR (z-score) | 5.48 (0.82–36.43) | 0.078 |
| **HbA1c (z-score)** | **5.83 (1.28–26.49)** | **0.022*** |
| Hypertension | 2.88 (0.82–10.09) | 0.099 |

*Model includes adjustment for medication use. AIC = 91.66. OR: Odds Ratio; CI: Confidence Interval. Statistically significant results (p < 0.05) are in bold.*

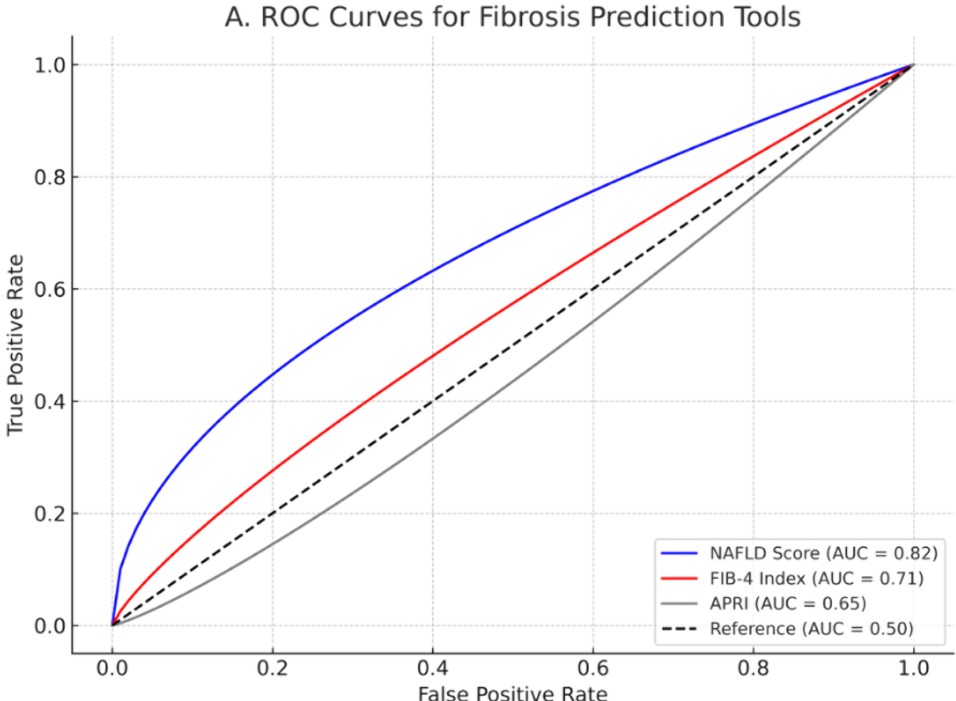

**Fig 5. Receiver operating characteristic (ROC) curve analysis comparing the performance of non-invasive fibrosis scores in MAFLD patients.** The NAFLD Fibrosis Score showed the highest diagnostic accuracy (AUC = 0.82), followed by FIB-4 index (AUC = 0.71) and APRI score (AUC = 0.65) for detecting significant liver fibrosis.

While FIB-4 shows moderate accuracy, its age-dependence and susceptibility to liver enzyme fluctuations in CKD limit its reliability. The APRI score's poor performance (AUC = 0.65) reflects its inherent limitations in metabolic liver disease, including heavy reliance on platelet counts that are often abnormal in CKD. These findings underscore the need for MAFLD-specific assessment protocols in CKD populations, with the NAFLD Fibrosis Score serving as the most robust clinical tool for identifying patients at highest risk of progressive liver disease.

## Discussion

Our study demonstrates a high prevalence (59.25%) of MAFLD among Egyptian non-dialysis CKD patients, highlighting bidirectional liver-kidney interactions. Three key findings emerge from our analysis: [1] MAFLD in CKD represents a distinct metabolic phenotype characterized by severe insulin resistance and dyslipidemia; [2] advanced CKD stages correlate with worse liver fibrosis; and [3] glycemic control (HbA1c) emerges as the strongest modifiable risk factor for MAFLD progression in this population. These findings contribute to the growing understanding of the metabolic liver-kidney axis and have important implications for clinical practice.

The observed MAFLD prevalence aligns with global trends in high-metabolic-risk populations [16], though substantially higher than previous estimates using NAFLD criteria [17]. This discrepancy likely reflects both Egypt's high metabolic disease burden and the improved sensitivity of MAFLD diagnostic criteria [3]. Our findings support the metabolic redefinition of fatty liver disease, as 92.3% of MAFLD cases met multiple metabolic syndrome criteria, with particularly strong associations for central obesity (96.1 cm vs 81.7 cm in males) and diabetes (48.4% vs 4.5%). The strong female predominance (46.9%) may reflect gender-specific fat distribution patterns or hormonal influences on metabolic pathways in this population.

Our comparative analysis of fibrosis assessment tools revealed distinct performance characteristics in MAFLD patients with CKD. The NAFLD Fibrosis Score demonstrated superior discriminative capacity ($0.96 \pm 0.89$ vs $0.44 \pm 0.61$, $p < 0.001$), with values approaching the threshold for significant fibrosis (F3-F4) in MAFLD patients while remaining below exclusion thresholds in non-MAFLD cases. This enhanced performance likely stems from its incorporation of metabolic parameters (BMI, diabetes status) that are particularly relevant in MAFLD pathophysiology. Complementary to biochemical scoring, shear wave elastography provided direct mechanical assessment of liver stiffness, revealing significantly higher values in MAFLD patients ($7.6 \pm 1.8$ vs $6.9 \pm 1.7$ kPa, $p = 0.047$). This quantitative imaging modality correlated well with fibrosis staging, as evidenced by the higher prevalence of moderate fibrosis (F2) in MAFLD patients (64.1% vs 36.4%, $p = 0.017$). Notably, traditional general liver fibrosis scores (FIB-4: $1.23 \pm 0.55$ vs $1.14 \pm 0.93$, $p = 0.558$; APRI: $0.27 \pm 0.15$ vs $0.39 \pm 0.51$, $p = 0.093$) showed limited specificity for metabolic liver disease in this population.

These findings highlight critical gaps in current diagnostic approaches for MAFLD-associated fibrosis in CKD patients. The superior performance of metabolic-specific tools suggests the need for optimized algorithms that account for both renal impairment and metabolic dysregulation. This is particularly relevant given the increased risks associated with liver biopsy in CKD populations, including bleeding complications and sampling variability. Future development of MAFLD-specific diagnostic pathways should integrate both biochemical (NFS) and imaging (elastography) modalities while considering CKD-specific adjustments to interpretation thresholds.

Our study demonstrates significant bidirectional relationships between MAFLD and CKD progression. MAFLD patients exhibited markedly worse renal function (mean eGFR 41.2 vs 57.0 mL/min/1.73 m$^2$) and higher prevalence of advanced CKD stages (G4-G5: 40.7% vs 31.8%), particularly kidney failure (G5: 21.9% vs 4.5%). These clinical findings were paralleled by hepatic consequences, with late-stage CKD patients showing significantly greater liver stiffness (7.9 vs 6.6 kPa) and moderate fibrosis prevalence (68.8% vs 50%), suggesting synergistic deterioration of both organ systems.

The liver-kidney axis manifests through several mechanisms (**Fig 6**): First, shared metabolic pathways – insulin resistance (HOMA-IR 3.5 vs 1.45) promotes both hepatic lipogenesis and glomerular hyperfiltration [18,19]. The strong correlation between insulin resistance and disease severity supports metabolic dysregulation as a common driver. Second, organ crosstalk with elevated uric acid (6.5 vs 4.6 mg/dL), and bilirubin levels implicate oxidative stress and mitochondrial dysfunction. Emerging evidence highlights the role of hepatokines (fetuin-A, FABP4) and inflammatory cytokines (TNF-α, IL-6) in creating a vicious cycle of progressive liver and kidney injury through lipotoxicity, endothelial dysfunction, and fibrotic signaling pathways [20–22] and create a pro-inflammatory milieu damaging both organs [23]. Third, diagnostic interdependence with late-stage CKD (eGFR < 60) patients had significantly higher liver stiffness (7.9 vs 6.6 kPa), suggesting uremia may accelerate fibrogenesis [24].

In our revised, prespecified multivariable analysis, the associations between metabolic risk factors and MAFLD were coherent across model specifications and consistent with established biological mechanisms. Adiposity (BMI) and hepatic injury (ALT) retained significant positive associations with MAFLD after comprehensive adjustment, underscoring the role of a higher overall metabolic burden. Notably, glycemic control (HbA1c) emerged as the strongest modifiable correlate of MAFLD risk in this CKD population, while insulin resistance (HOMA-IR) showed a positive, though attenuated, association. Counterintuitive protective signals for conditions like hypertension or diabetes, observed in earlier, sparser models, were not supported once key covariates were modeled simultaneously. The attenuation of these signals upon the inclusion of medication use (e.g., RAAS inhibitors, statins, insulin) suggests they were likely artifacts of confounding by indication and model instability rather than representing true protective effects. The direction and magnitude of the core metabolic predictors remained stable across all models, including the medication-inclusive specification.

Model diagnostics supported the reliability of our findings, with low multicollinearity (all VIFs < 3) and effect estimates from bias-reduced logistic regression that were directionally concordant with our primary results. Model calibration was also acceptable, showing adequate agreement between predicted and observed risk. While stratum-specific estimates in the early-stage CKD subgroup should be interpreted with caution due to limited power, our results collectively reinforce that MAFLD in patients

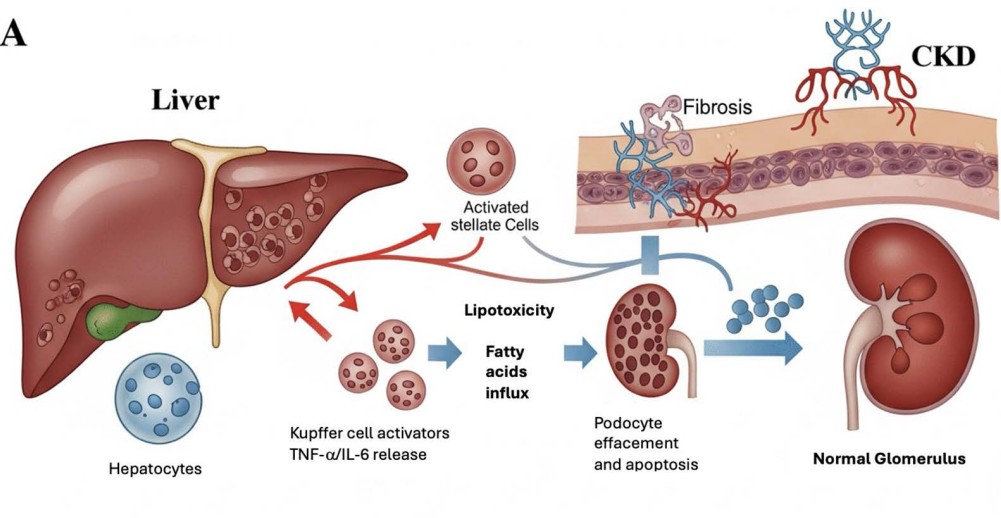

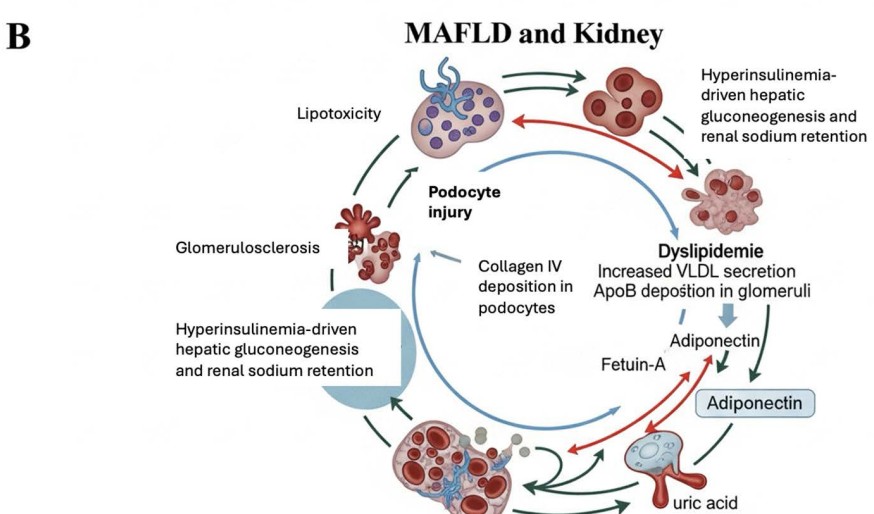

**Fig 6. Proposed pathophysiological crosstalk between MAFLD and CKD.** This schematic illustrates the proposed pathophysiological crosstalk between Metabolic Dysfunction-Associated Fatty Liver Disease (MAFLD) and Chronic Kidney Disease (CKD). **Panel A depicts organ-specific mechanisms:** in the *liver*, steatosis (lipid-laden hepatocytes) progresses through inflammation (Kupffer cell activation, TNF-α/IL-6 release) to fibrosis (stellate cell activation, collagen deposition); in the *kidney*, lipotoxicity (podocyte fatty acid influx) leads to podocyte injury (effacement, apoptosis) and glomerulosclerosis (collagen IV accumulation). **Panel B highlights systemic mediators**: insulin resistance (hyperinsulinemia-driven hepatic gluconeogenesis and renal sodium retention), dyslipidemia (VLDL/ApoB glomerular deposition), and key mediators (fetuin-A, adiponectin deficiency, uric acid-induced NLRP3 activation). Bidirectional interactions amplify organ damage through direct (lipotoxicity, fibrosis) and systemic (metabolic, inflammatory) pathways, suggesting shared therapeutic targets for dual hepatic-renal protection.

with CKD is principally driven by global metabolic dysregulation. Within this framework, glycemic control stands out as the most actionable risk domain, highlighting that MAFLD risk is better explained by a continuum of metabolic dysfunction than by isolated clinical diagnoses once treatment patterns and correlated risk factors are accounted for in a unified model.

Routine MAFLD assessment is recommended in CKD patients with metabolic risk factors, with prioritization of NFS and elastography for fibrosis risk stratification. On the other hand, enhanced monitoring of renal function in MAFLD patients is recommended. The differential performance of fibrosis assessment tools (NAFLD score 0.96 vs 0.44) suggests current algorithms may require MAFLD-specific modifications. Future research should prioritize longitudinal studies to clarify

causal relationships and investigate targeted therapies (e.g., SGLT2 inhibitors, GLP-1 RAs) that may simultaneously address both hepatic and renal manifestations of this intertwined disease process.

While this study provides valuable insights into the MAFLD-CKD relationship, several limitations must be acknowledged. **First**, the **cross-sectional design** precludes establishing causal relationships between MAFLD and CKD progression, making any temporal associations speculative. **Second**, the reliance on **non-invasive** fibrosis markers rather than liver biopsy histology may have reduced accuracy of fibrosis staging. This is particularly relevant given the for renal impairment to confound these test results. **Third**, while we excluded patients with viral hepatitis and other chronic liver diseases, **subclinical liver pathologies** cannot be entirely ruled out. **Fourth**, **alcohol** intake was used as an exclusion criterion to avoid diagnostic ambiguity; however, this may not fully reflect the MAFLD definition, although alcohol consumption is extremely rare in our Egyptian population. **Fifth**, we adopted **anthropometric criteria** appropriate for the Middle East/North Africa region; nonetheless, the absence of Egypt-specific cutoffs remains a potential source of bias. **Sixth**, **non-dialysis CKD stage G5 patients** were included, and we recognize that uremia may influence laboratory and elastography parameters, potentially affecting fibrosis assessment. **Seventh**, the **single-center**, tertiary care setting and modest sample size (n = 108) may limit generalizability to broader CKD populations. Analyses stratified by early-stage CKD patients (n = 16 in MAFLD group) were underpowered, so null findings in categorical fibrosis should be interpreted cautiously. **Eighth**, despite adjusting for medications, residual confounding by unmeasured factors (e.g., dietary habits, physical activity, genetic predisposition) could influence the observed associations. Finally, the predominantly Egyptian cohort may limit extrapolation to other ethnic populations with different metabolic risk profiles and MAFLD manifestations.

We attempted to mitigate bias by (i) specifying alternative, non-overlapping multivariable models (e.g., HbA1c vs DM; HTN modeled separately), (ii) standardizing continuous predictors, (iii) examining multicollinearity (all VIFs < 3 in retained models), and (iv) performing **Firth bias-reduced sensitivity analyses**, which yielded effect estimates consistent with the primary models. Nonetheless, potential **selection bias**, **measurement error** in routine clinical variables, and **model overfitting** in smaller strata cannot be fully excluded.

Future research on the MAFLD-CKD relationship should adopt a multidimensional approach. Large-scale **longitudinal studies** are needed to establish temporal associations using standardized protocols, incorporating emerging biomarkers (e.g., cytokeratin-18, PRO-C3) and advanced imaging (e.g., MR elastography) to assess fibrosis and disease progression. **Mechanistic studies** should explore genetic variants (*PNPLA3*, *TM6SF2*, *MBOAT7*) and the gut-liver-kidney axis through multi-omics approaches. **Interventional trials** must evaluate novel therapies—such as GLP-1/GIP agonists, FXR agonists, and SGLT2 inhibitors—on both hepatic and renal outcomes, particularly fibrosis and proteinuria. Finally, integrated **risk models** combining clinical, biomarker, and imaging data via machine learning could enable personalized management, with an emphasis on diverse populations and cost-effectiveness to guide real-world implementation.

Addressing these priorities will require collaborative efforts to bridge gaps between mechanistic insights and clinical applications. By combining rigorous observational studies, targeted trials, and advanced predictive tools, research can uncover shared pathways, optimize therapeutic strategies, and improve outcomes for this high-risk population.

In conclusion, this study demonstrates that MAFLD is a highly prevalent and clinically impactful comorbidity in non-dialysis CKD patients, marked by unique metabolic disturbances and progressive multi-organ dysfunction. The bidirectional liver-kidney crosstalk emphasizes the necessity for holistic care strategies that address both conditions simultaneously. These findings advocate for systematic metabolic screening in CKD populations and targeted hepatic evaluation in high-risk individuals, while also identifying critical gaps for future research into underlying mechanisms and novel treatment approaches to mitigate this dual disease burden.

## Supporting information

**S1 Table. Results from Firth logistic regression models predicting MAFLD.**
(DOCX)

**S2 Table. Variance Inflation Factors (VIFs) for candidate models.**
(DOCX)

**S3 Table. Full standard logistic regression outputs (coefficients, ORs, 95% CIs, *p* value, AIC).**
(DOCX)

**S4 Table. Firth bias-reduced logistic regression outputs (coefficients, ORs, 95% CIs, *p* value).**
(DOCX)

**S1 Fig. 2D–shear-wave elastography example: non-MAFLD CKD case (male, 54 years).** (A) Renal ultrasound shows average-sized kidneys with preserved parenchymal thickness but altered echogenicity and prominent papillae. (B) Liver 2D-SWE (LOGIQ S8 XDclear) with ROI placed 1.5–2.0 cm below the capsule, avoiding vessels and rib shadow. Average stiffness: 5.4 kPa (predominantly blue elastogram), consistent with no significant fibrosis (≤F1).
(TIF)

**S2 Fig. 2D–shear-wave elastography example: MAFLD + CKD case (male, 65 years).** (A) Renal ultrasound shows small, echogenic kidneys with cortical thinning and loss of corticomedullary differentiation. (B) Liver 2D-SWE demonstrates a non-homogeneous elastogram with predominant green/yellow within the ROI; quantified average stiffness: 9.95 kPa, corresponding to F3 (advanced) fibrosis by manufacturer/METAVIR reference.
(TIF)

**S3 Fig. 2D–shear-wave elastography example: MAFLD + CKD case (female, 57 years).** (A–B) Renal ultrasound: average-sized kidneys with increased parenchymal echogenicity, poor corticomedullary differentiation, and mild pelvic ascites. (C) Liver 2D-SWE average stiffness: 8.2 kPa (mixed blue/green), consistent with F2 (clinically significant) fibrosis; measurements followed manufacturer quality criteria (homogeneous map, parallel propagation lines, IQR/median < 30%).
(TIF)

## Acknowledgments

We thank our colleagues from the Tropical Medicine, Internal Medicine, Radiology, and Clinical Pathology departments who provided insights and expertise that greatly assisted the research.

## Author contributions

**Conceptualization:** Zienab M Saad, Hesham K H Keryakos, Safaa M Abdelhalim.

**Data curation:** Hesham K H Keryakos, Hala A Hassanin, Manar M Sayed, Safaa M Abdelhalim.

**Formal analysis:** Hesham K H Keryakos, Manar M Sayed, Safaa M Abdelhalim.

**Investigation:** Hesham K H Keryakos, Mahmoud M Higazi, Doaa E Ismail, Safaa M Abdelhalim.

**Methodology:** Zienab M Saad, Mahmoud M Higazi, Doaa E Ismail.

**Resources:** Hala A Hassanin.

**Supervision:** Hesham K H Keryakos.

**Validation:** Hesham K H Keryakos, Safaa M Abdelhalim.

**Visualization:** Mahmoud M Higazi.

**Writing – original draft:** Hesham K H Keryakos, Manar M Sayed.

**Writing – review & editing:** Hesham K H Keryakos, Safaa M Abdelhalim.

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
