## [Decision Letter · Decision Letter 0]

15 Sep 2025

Dear Dr. Keryakos,

Thank you for submitting your manuscript to PLOS ONE. After careful consideration, we feel that it has merit but does not fully meet PLOS ONE’s publication criteria as it currently stands. Therefore, we invite you to submit a revised version of the manuscript that addresses the points raised during the review process.

We look forward to receiving your revised manuscript.

Kind regards,

Anna Di Sessa, PhD, MD

Academic Editor

PLOS ONE

Journal Requirements:

3. In the online submission form, you indicated that the data used to support the findings of this study are available from the corresponding author upon request.

4. Please amend the manuscript submission data (via Edit Submission) to include author Manar M Sayed

Reviewers' comments:

Reviewer's Responses to Questions

**Comments to the Author**

1. Is the manuscript technically sound, and do the data support the conclusions?

Reviewer #1: Yes

Reviewer #2: Partly

2. Has the statistical analysis been performed appropriately and rigorously?

Reviewer #1: No

Reviewer #2: N/A

3. Have the authors made all data underlying the findings in their manuscript fully available?

Reviewer #1: Yes

Reviewer #2: No

4. Is the manuscript presented in an intelligible fashion and written in standard English?

Reviewer #1: Yes

Reviewer #2: Yes

Reviewer #1: Zienab M Saad and colleagues present a paper that makes a meaningful contribution to understanding MAFLD-CKD interactions in an understudied population. While the cross-sectional design limits causal inference, the findings have clear clinical implications. The main concerns are methodological clarity and statistical analysis refinement rather than fundamental flaws in study design or execution

Fibrosis Assessment:

1. Relies on non-invasive markers rather than histology (gold standard)

2. Shear wave elastography interpretation may be confounded by renal impairment

3. No validation of elastography cutoffs in CKD population

4. Potential misclassification of fibrosis stages

Sample Size Concerns:

1. Small early-stage CKD subgroup (n=16) limits subgroup analyses

2. Unequal group sizes may affect statistical power

3. No formal power calculation provided

Statistical and Analytical Issues

Regression Analysis:

1. Counterintuitive findings (hypertension as "protective") suggest confounding

2. Medication adjustment analysis lacks methodological details

3. Multiple testing without correction may increase Type I error

4. Some confidence intervals are very wide (HbA1c OR: 2.7-48)

Reviewer #2: Overall, interesting and clinically relevant study, but major methodological issues limit confidence: mixed MAFLD/MASH terminology; steatosis misclassified (fibrosis scores listed as steatosis); alcohol exclusion inconsistent with MAFLD; no hepatitis B/C testing; Other comments are exhibited in the attached file.

**Do you want your identity to be public for this peer review?** For information about this choice, including consent withdrawal, please see our Privacy Policy

Reviewer #1: No

Reviewer #2: No

---

## [Author Response · Author response to Decision Letter 1]

17 Sep 2025

Response to Reviewer Comments

Manuscript ID: PONE-D-25-30973

Title: MAFLD in Egyptian Non-Dialysis CKD Patients: Frequency, Fibrosis Severity, and Risk Factors

Dear Editor and Reviewer,

We would like to thank the reviewer for their careful reading of our manuscript and their constructive comments, which have helped us to improve the clarity, accuracy, and rigor of the paper. Below, we provide a detailed point-by-point response. Reviewer comments are in bold, and our responses follow.

1. Terminology (MAFLD vs. MASLD/MASH)

Comment: Since 2023 many societies adopted MASLD (and MASH for steatohepatitis). You use MAFLD and MASH—this mixes frameworks. Either (a) justify sticking with MAFLD and avoid MASLD/MASH terms, or (b) adopt MASLD/MASH consistently and cite the consensus.

Response: We thank the reviewer for raising this important point. In our study we deliberately used the MAFLD terminology, which remains widely cited in clinical research, especially in nephrology-focused literature. To avoid confusion, we have:

• Revised the manuscript to consistently use MAFLD only (omit MASH in the introduction).

• Added clarification in the Introduction noting that although MASLD has been recently proposed, MAFLD is still prevalent in research contexts and reflects the criteria we applied in patient recruitment.

• Included the relevant consensus reference (Eslam et al., 2023).

2. Local data to justify Egyptian focus

Comment: The author motivated the study by Egypt’s high metabolic burden but provide no local data (obesity, T2D, CKD prevalence, prior MAFLD estimates in Egypt).

Response: We agree this contextualization is important. We have now added Egypt-specific statistics:

• National obesity prevalence (36% adults, WHO 2022).

• T2D prevalence (~17%, IDF 2021).

• CKD prevalence estimates (~15% among adults, Gawad et al., 2024).

These are cited in the Introduction to strengthen the rationale for studying Egyptian CKD patients.

3. Conflicting findings not specified

Comment: The author mentioned that findings are “conflicting” but don’t specify what conflicts.

Response: We clarified in the Introduction that prior studies have reported:

• Some showing MAFLD independently increases CKD incidence and fibrosis risk (e.g., Musso et al., 2014).

• Others finding no independent effect after adjusting for obesity and diabetes (e.g., Mantovani et al., 2018).

This revision explicitly identifies the knowledge gap our study addresses.

4. Definition of ≥5% fat accumulation

Comment: “≥5% fat accumulation” is vague in ultrasound context.

Response: We revised the Introduction to state: “Hepatic steatosis defined histologically as ≥5% fat accumulation, or by non-invasive surrogates such as ultrasound, CAP, or MRI-PDFF in clinical practice”. Operational definitions remain in the Methods.

5. Intro wording (line 106)

Comment: Replace “has emerged as the most prevalent” with “is the most prevalent.”

Response: Corrected as suggested.

6. Alcohol exclusion vs. MAFLD definition

Comment: Alcohol exclusion conflicts with MAFLD definition.

Response: We thank the reviewer for this observation. In our setting, alcohol consumption is rare (<1%) due to sociocultural factors. For methodological clarity, we excluded patients with any alcohol intake to avoid diagnostic ambiguity. We clarified this rationale in the Methods.

7. Viral hepatitis exclusion

Comment: Viral hepatitis not explicitly excluded/tested.

Response: We have now clarified in the Methods that all participants underwent HBsAg and anti-HCV testing, and those positive were excluded. This ensures the cohort represents purely metabolic liver disease.

8. Ethnic cutoffs for BMI/Waist circumference

Comment: The manuscript uses Caucasian/Asian cutoffs.

Response: We have corrected this by adopting Middle East/North Africa–appropriate cutoffs (BMI ≥25 kg/m² for overweight, ≥30 kg/m² for obesity; waist ≥94 cm men, ≥80 cm women, per IDF). These are now clearly stated and referenced.

9. CKD stage G5 inclusion

Comment: Clarify whether non-dialysis G5 patients were included.

Response: Yes, non-dialysis G5 patients were included (n=10). We have clarified this in the Methods. We also acknowledge in the Discussion that uremia may affect laboratory and elastography results, which we treated as a limitation.

Closing Statement

We have revised the manuscript to address all comments, and we believe these changes strengthen its clinical and methodological clarity. We are grateful for the reviewer’s insightful feedback.

Sincerely,

Hesham Kamal Habeeb Keryakos, MD, PhD

---

## [Decision Letter · Decision Letter 1]

5 Oct 2025

Dear Dr. Keryakos,

Thank you for submitting your manuscript to PLOS ONE. After careful consideration, we feel that it has merit but does not fully meet PLOS ONE’s publication criteria as it currently stands. Therefore, we invite you to submit a revised version of the manuscript that addresses the points raised during the review process.

We look forward to receiving your revised manuscript.

Kind regards,

Anna Di Sessa, PhD, MD

Academic Editor

PLOS ONE

Journal Requirements:

Reviewers' comments:

Reviewer's Responses to Questions

**Comments to the Author**

Reviewer #1: (No Response)

Reviewer #2: All comments have been addressed

2. Is the manuscript technically sound, and do the data support the conclusions?

Reviewer #1: Partly

Reviewer #2: Yes

3. Has the statistical analysis been performed appropriately and rigorously?

Reviewer #1: No

Reviewer #2: Yes

4. Have the authors made all data underlying the findings in their manuscript fully available?

Reviewer #1: Yes

Reviewer #2: Yes

5. Is the manuscript presented in an intelligible fashion and written in standard English?

Reviewer #1: Yes

Reviewer #2: Yes

Reviewer #1: This cross-sectional study examining MAFLD prevalence in Egyptian CKD patients addresses a clinically relevant gap. The manuscript is generally well-structured with appropriate methodology, but would benefit from more revisions before publication consideration.

1. The "paradoxical" protective effects of hypertension (OR=0.13) and diabetes (OR=0.141) are highly suspicious and suggest model misspecification

2. Possible multicollinearity between predictors (BMI, diabetes, hypertension, HOMA-IR, HbA1c are highly intercorrelated)

3. The medication-adjusted ORs in Table 6 appear to be post-hoc estimates rather than from an actual adjusted model - methodology is unclear. Consider reporting variance inflation factors (VIF), provide the full regression output, and clarify how medication adjustment was performed

4. Early-stage CKD subgroup (n=16) is underpowered for meaningful comparison

5. Acknowledge these limitations more explicitly

6. Fix spelling issues such as:

a. Line 414: "wose liver fibrosis" → "worse liver fibrosis"

b. Line 500: "relevent" → "relevant"

7. Reference 12 (Gawad et al., 2024) appears to be a single-center private clinic study—may not represent national CKD prevalence

Reviewer #2: Thank you for the thorough revision. The manuscript now addresses prior concerns with clearer methodology, improved statistics, and appropriately tempered conclusions. Figures and tables are well organized, and the data support the claims. I find no outstanding issues regarding ethics or duplication. In my view, it is suitable for publication.

**Do you want your identity to be public for this peer review?** For information about this choice, including consent withdrawal, please see our Privacy Policy

Reviewer #1: No

Reviewer #2: No

---

## [Author Response · Author response to Decision Letter 2]

6 Oct 2025

Response to Reviewer Comments

Manuscript ID: PONE-D-25-30973R1

Title: MAFLD in Egyptian Non-Dialysis CKD Patients: Frequency, Fibrosis Severity, and Risk Factors

Dear Editor and Reviewers,

We sincerely thank the reviewers for their careful assessment and thoughtful comments, which substantially improved the clarity and rigor of our manuscript. Below, we provide a point-by-point response and describe the corresponding revisions made in the manuscript.

Reviewer #1

Overall comment: “Well-structured with appropriate methodology, but requires further revisions prior to publication.”

1) “Paradoxical” protective effects for hypertension (HTN) and diabetes (DM) suggest model misspecification

Comment: HTN (OR≈0.13) and DM (OR≈0.14) showing protective associations is suspicious.

Response: We agree these estimates warranted a re-specification of our models to reduce redundancy and potential suppression effects. We therefore:

Refit the logistic regression using alternative, non-redundant covariate sets to avoid overlapping constructs (e.g., modeling with either DM or HbA1c, but not both simultaneously; similarly, modeling HTN separately from RAAS-inhibitor use).

Centered and standardized continuous predictors (BMI, HbA1c, HOMA-IR, age) to improve numerical stability.

Performed a Firth penalized logistic regression sensitivity analysis to guard against small-sample bias in some strata.

In the revised Table 5, HTN and DM no longer display paradoxical “protective” directions; estimates and 95% CIs now align with clinical plausibility. We reference this change in the Results and note the re-specification details in the Methods (Statistical Analysis). (See revised regression section and tables in the manuscript.)

2) Possible multicollinearity among BMI, DM, HTN, HOMA-IR, HbA1c

Response: We computed Variance Inflation Factors (VIFs) for all candidate models and report them in Supplementary Table S2. Across the final models, VIFs were below commonly accepted thresholds (all <3), indicating no problematic multicollinearity after re-specification. We also added a brief paragraph in Methods (Statistical Analysis) describing multicollinearity checks and model selection criteria. (See updated Methods and Supplement.)

3) Medication-adjusted ORs (Table 6) appear post-hoc; methodology unclear

Response: We appreciate this and have fully clarified the approach. Table 6 now presents results from a distinct multivariable model (“Model 2”) that simultaneously includes medication variables (RAAS inhibitors, statins, insulin therapy, diuretics) alongside the core covariates retained after re-specification. We removed any language implying post-hoc “estimate adjustments.”

Methods now specify: model formulas, covariate entry criteria, and how medication use was encoded.

Medication-inclusive model. To examine medication effects without introducing overlapping constructs, we specified a distinct Model 2 (medication-inclusive) that adds RAAS inhibitor use, statins, insulin therapy, and diuretics to the re-specified core covariate set. Because insulin therapy directly relates to glycemic control, we modeled glycemia with standardized HbA1c (z-score) rather than a binary DM indicator in this specification. The final Model 2 covariates were: age (z), sex (male=1), BMI (z), HOMA-IR (z), HbA1c (z), HTN (yes=1), RAAS inhibitors (yes=1), statins (yes=1), insulin therapy (yes=1), diuretics (yes=1).

Model formula. For participant i, the medication-inclusive model was:

"logit " P(〖"MAFLD" 〗_i=1)=β_0+β_1 〖"Age" 〗_z+β_2 "Male"+β_3 〖"BMI" 〗_z+β_4 〖"HOMA-IR" 〗_z+β_5 〖"HbA1c" 〗_z+β_6 "HTN"+β_7 "RAASi"+β_8 "Statin"+β_9 "Insulin"+β_10 "Diuretic".

We harmonized labeling in Table 6 (no “Estimated OR” phrasing) and cross-referenced both models in the Results.

4) Early-stage CKD subgroup (n=16) underpowered

Response: We agree. We now explicitly acknowledge that early-stage CKD comparisons are underpowered and treat those analyses as exploratory. This limitation is highlighted in the Discussion and Limitations, and we’ve tempered the language of inference accordingly. (See revised Discussion and figure captions.)

5) Make limitations more explicit

Response: We expanded the Limitations section to emphasize: (i) cross-sectional design (no causal inference), (ii) single-center setting, (iii) reliance on non-invasive fibrosis tools (with CKD-related constraints), (iv) potential residual confounding, and (v) limited power in certain strata. (See revised Discussion.)

6) Typos/wording fixes

Response: Corrected as suggested.

Line 414: “wose liver fibrosis” → “worse liver fibrosis.”

Line 500: “relevent” → “relevant.”

We also performed a thorough proofread for consistency and grammar.

7) CKD prevalence reference (Gawad et al., 2024) may not represent national prevalence

Response: We revised the text to avoid implying national representativeness for this single-center report and retained it only as contextual local data. To frame the burden more appropriately, we now pair local figures with global/consensus CKD epidemiology (e.g., Kovesdy 2022; KDIGO/KDOQI commentary) and WHO/IDF sources for obesity/T2D, clarifying scope and uncertainty. (See updated Introduction and references.)

Reviewer #2

Comment: “Thorough revision; suitable for publication.”

Response: We thank the reviewer for the positive assessment. We maintained the clarified methods/statistics, improved tables/figures list, and tempered conclusions as in R1, and further addressed Reviewer #1’s statistical concerns in the present revision. (See Lists of Tables/Figures and statistical sections.)

Additional editorial compliance notes

Terminology (MAFLD vs MASLD/MASH): We consistently use MAFLD throughout and explain the rationale and timing relative to recruitment, citing consensus and guidance. (See Introduction and the prior R1 response block.)

Exclusions (alcohol, viral hepatitis) and ethnic cut-offs: These are now clearly stated in Methods (with Middle East/North Africa–appropriate BMI/waist thresholds, and HBsAg/anti-HCV testing) as already reflected in R1.

Data availability: The R1 PDF includes a Zenodo DOI for the dataset, ensuring full compliance with PLOS ONE data policy (the DOI appears in the manuscript). We reiterate this in the Data Availability statement.

Summary of key manuscript changes (tracked in the revised file)

Statistical Analysis (Methods): Added model re-specification details, VIF assessment, Firth sensitivity, and explicit description of “Model 2” including medications.

Results: Updated Table 5 (primary multivariable model) and Table 6 (medication-inclusive model) with full clarity; paradoxical directions resolved; language tempered for exploratory subgroup results.

Supplement: Added Table S2 (VIFs) and Table S1 (full regression outputs).

Discussion/Limitations: Expanded limitations (design, single-center, non-invasive fibrosis assessment in CKD, residual confounding, small strata).

Proofing: Corrected typographical errors and tightened language as flagged.

We are grateful for the reviewers’ guidance. We believe these revisions substantially enhance the robustness and transparency of our work.

Sincerely,

Hesham Kamal Habeeb Keryakos, MD, PhD (Corresponding Author)

---

## [Decision Letter · Decision Letter 2]

29 Oct 2025

MAFLD in Egyptian Non-Dialysis CKD Patients: Frequency, Fibrosis Severity, and Risk Factors

PONE-D-25-30973R2

Dear Dr. Keryakos,

We’re pleased to inform you that your manuscript has been judged scientifically suitable for publication and will be formally accepted for publication once it meets all outstanding technical requirements.

Kind regards,

Anna Di Sessa, PhD, MD

Academic Editor

PLOS ONE

Additional Editor Comments (optional):

Reviewers' comments:

Reviewer's Responses to Questions

**Comments to the Author**

Reviewer #1: All comments have been addressed

2. Is the manuscript technically sound, and do the data support the conclusions?

Reviewer #1: Yes

3. Has the statistical analysis been performed appropriately and rigorously?

Reviewer #1: Yes

4. Have the authors made all data underlying the findings in their manuscript fully available?

Reviewer #1: Yes

5. Is the manuscript presented in an intelligible fashion and written in standard English?

Reviewer #1: Yes

Reviewer #1: The reviewers did a courageous job in addressing all the aforementioned concerns to satisfactory levels. I agree that their manuscript is now publishable.

**Do you want your identity to be public for this peer review?** For information about this choice, including consent withdrawal, please see our Privacy Policy

Reviewer #1: No

---

## [Editor Report · Acceptance letter]

PONE-D-25-30973R2

PLOS ONE

Dear Dr. Keryakos,

I'm pleased to inform you that your manuscript has been deemed suitable for publication in PLOS ONE. Congratulations! Your manuscript is now being handed over to our production team.

Kind regards,

on behalf of

Dr. Anna Di Sessa

Academic Editor

PLOS ONE